# Silicon Amendment Enhances Agronomic Efficiency of Nitrogen Fertilization in Maize and Wheat Crops under Tropical Conditions

**DOI:** 10.3390/plants10071329

**Published:** 2021-06-29

**Authors:** Fernando Shintate Galindo, Paulo Humberto Pagliari, Willian Lima Rodrigues, Guilherme Carlos Fernandes, Eduardo Henrique Marcandalli Boleta, José Mateus Kondo Santini, Arshad Jalal, Salatiér Buzetti, José Lavres, Marcelo Carvalho Minhoto Teixeira Filho

**Affiliations:** 1Center for Nuclear Energy in Agriculture (CENA), University of São Paulo (USP), Piracicaba 13416-000, Brazil; 2Department of Soil, Water, and Climate, Southwest Research and Outreach Center (SWROC), University of Minnesota (UMN), Lamberton, MN 56152, USA; pagli005@umn.edu; 3Department of Plant Health, Rural Engineering, and Soils (DEFERS), São Paulo State University (UNESP), Ilha Solteira 15345-000, Brazil; willianrodrigues53@gmail.com (W.L.R.); guilherme.carlos.fernandes@gmail.com (G.C.F.); eduardomarcandalli7@gmail.com (E.H.M.B.); santinijmk@gmail.com (J.M.K.S.); arshad.jalal@unesp.br (A.J.); sbuzetti@agr.feis.unesp.br (S.B.); jlavres@usp.br (J.L.); mcm.teixeira-filho@unesp.br (M.C.M.T.F.)

**Keywords:** Silicon application, sustainable crop production, tropical agriculture, Ca and Mg silicate, improved nitrogen management

## Abstract

Sustainable management strategies are needed to improve agronomic efficiency and cereal yield production under harsh abiotic climatic conditions such as in tropical Savannah. Under these environments, field-grown crops are usually exposed to drought and high temperature conditions. Silicon (Si) application could be a useful and sustainable strategy to enhance agronomic N use efficiency, leading to better cereal development. This study was developed to explore the effect of Si application as a soil amendment source (Ca and Mg silicate) associated with N levels applied in a side-dressing (control, low, medium and high N levels) on maize and wheat development, N uptake, agronomic efficiency and grain yield. The field experiments were carried out during four cropping seasons, using two soil amendment sources (Ca and Mg silicate and dolomitic limestone) and four N levels (0, 50, 100 and 200 kg N ha^−1^). The following evaluations were performed in maize and wheat crops: the shoots and roots biomass, total N, N-NO_3_^−^, N-NH_4_^+^ and Si accumulation in the shoots, roots and grain tissue, leaf chlorophyll index, grain yield and agronomic efficiency. The silicon amendment application enhanced leaf chlorophyll index, agronomic efficiency and N-uptake in maize and wheat plants, benefiting shoots and roots development and leading to a higher grain yield (an increase of 5.2 and 7.6%, respectively). It would be possible to reduce N fertilization in maize from 185–180 to 100 kg N ha^−1^ while maintaining similar grain yield with Si application. Additionally, Si application would reduce N fertilization in wheat from 195–200 to 100 kg N ha^−1^. Silicon application could be a key technology for improving plant-soil N-management, especially in Si accumulator crops, leading to a more sustainable cereal production under tropical conditions.

## 1. Introduction

Silicon (Si) is the second most abundant element after oxygen in the Earth’s crust, comprising approximately 29% (28.8% wt) of the Earth’s crust [1,2,3]. Although Si is considered a quasi-essential element rather than a plant nutrient, it is increasingly being applied in agricultural systems worldwide [2,3,4,5]. Silicon content in the soil ranges from almost 1 to 45% depending on soil types [6]. The potential of Si in improving crop yield has been demonstrated in many studies, especially under biotic (pathogens attack) and abiotic (e.g., salinity, drought, high temperature, heavy metals toxicity) stress conditions [7,8]. Silicon is known for its role in alleviating negative effects of stress on many plant species. Monocotyledons in general and *Poaceae* species such as maize *(Zea mays* L.) and wheat (*Triticum aestivum* L.) are clearly favored due to an enhanced supply of Si [3,6,9]. It has been reported that some *Poaceae* species could accumulate Si to a level above 1% of total shoot biomass [7].

The cultivation and harvest of Si-accumulating crops is responsible for a constant depletion of the Si reservoir in soils [10,11] which may decrease Si bioavailability [12]. The enhanced Si removal by Si-accumulating crops, therefore, disrupts the recycling of Si by plants back into the soil [13]. Liang et al. [14] reported that Si content in Oxisols in the tropical zone can be less than 1% due to intense weathering processes. Highly weathered tropical and subtropical soils under continuous cropping systems are generally low in available Si content due to heavy desilication of primary silicate minerals, release and leaching of basic cations with decreased base saturation, and crop removal [3,7,15,16]. Therefore, the decrease in Si availability in tropical soils might have significant impacts on cropping systems if not properly managed.

A further problem in weathered and acidic soils is related to the occurrence of high levels of exchangeable Al^3+^. Soil acidity and related Al^3+^ toxicity have long been recognized as significant and alarming constraints in cereal production under tropical conditions. Nonetheless, Ca and Mg silicate could be a viable alternative to increase Si availability and neutralize Al toxicity as an alternative to conventional liming. Silicate application could reduce the time associated with liming reactions in the soil profile due to their higher solubility (6–7 times more soluble than carbonates) and alkalinity compared with limestone [17,18]. However, most studies related to Si-amendment sources application focused on soil chemical shifts instead of plant growth promotion, and nutrient uptake and imbalance. The application of various forms of Si (i.e., slag materials, silica powder, silicates, among others) could improve plant growth and development with greater nutrient uptake in different crops [16,19,20,21]. Haddad et al. [22] indicated a beneficial effect of Si treatment in alleviating damage associated with nitrogen (N) deficiency in rapeseed (*Brassica napus* L.). These authors showed that Si supply modulates the root expression of a large panel of genes [23], promotes a stronger N uptake associated with the induction of root nitrate transporters, and delays leaf senescence in plants cultivated under N deficiency [22]. Nonetheless, Si application has been reported to increase N availability in the soil by modifying physicochemical (e.g., soil exchange capacity) [24,25] and biological properties of soil (e.g., increasing the biomass of microbial N fixers) [26]. Singh et al. [27] reported that Si application in rice (*Oryza sativa* L.) enhances N availability in soil and leads to an increase in N uptake by plants. Laîné et al. [28] concluded that Si application (12 kg Si ha^−1^ as silicic acid) may have increased N availability in the soil, improved N uptake, and led to an accumulation of N in rapeseed plants cultivated with high N rates (160 kg ha^−1^). In this regard, Si application could affect N availability to plants and enhance nutrient use efficiency (also known as agronomic efficiency), expressed as grain produced per unit of N applied. Thereby, Si fertilization might improve N fertilization management while reducing the need for mineral N fertilization in agricultural systems [19].

Nitrogen is considered one of the most limiting nutrients for crop growth and yield [29]. However, excessive N supply could directly contribute to soil acidification, ammonia (NH_3_) and nitrogen oxide (N_2_O, NO_2_ and NO) emissions, and N leaching with extended consequences on global warming [30]. The application of N to maize and wheat using a side-dressing approach in the Brazilian Savannah has led to an increase of about 15% in the farmer’s total operational profit [31,32]. Over the last decades, several management practices have been developed to enable farmers to reduce the application of N-based fertilizers in cereal crops in order to limit potential negative environmental impacts and cost resulting from the heavy application of chemical N fertilizer [33]. Integrating N management practices could contribute significantly to improved cereal production under the savannah conditions, particularly in a maize-wheat rotation, which has high N nutritional demands [34,35]. Moreover, developing management practices that enhance agronomic efficiency leading to a greater grain yield is required to foment sustainable agricultural practices in the tropical agriculture.

Therefore, we propose a novel approach to investigate the effect of Si-amendment application, combined with N levels in a maize-wheat cropping system under tropical conditions. This research could provide new clues on how Si application could be included as an important strategy for improving the sustainability of cereal production with an improved N fertilization management. We hypothesized that Si application would improve the agronomic efficiency and N accumulation in maize and wheat crops. The increased agronomic efficiency and N accumulation may provide increased shoot and root development which may lead to greater grain yield when compared to plots that did not receive Si application. Therefore, the objective of this study was to investigate the combined effects of Si-amendment source (Ca and Mg silicate) and N levels (control, low, medium and high levels) application on: (1) leaf chlorophyll index (LCI), inorganic (N-NH_4_^+^ and N-NO_3_^−^) and total N and Si content in maize and wheat plants; (2) plant biomass, grain yield and agronomic efficiency of individual treatments.

## 2. Results

### 2.1. Summary of the Statistical Analysis: Maize

Silicon application significantly affected LCI, root Si, shoot N-NO_3_^−^, shoot N-NH_4_^+^ and shoot Si accumulation, shoot biomass and grain yield (Appendix A, Table 1). Nitrogen application significantly affected LCI, root N, root Si, shoot N-NO_3_^−^, shoot N-NH_4_^+^ and shoot Si accumulation, shoot biomass and grain yield (Appendix A, Table 2). Years of study significantly affected root N-NO_3_^−^, root N, shoot N-NH_4_^+^, shoot N and shoot Si accumulation, root biomass, shoot biomass and grain yield (Appendix A, Table 3).

The interaction between silicon application and N levels affected root N-NO_3_^−^, root N-NH_4_^+^, shoot N, grain N and grain Si accumulation, root biomass and agronomic efficiency (Appendix A, Figure 1). The interaction between silicon application and years of study significantly affected root N-NH_4_^+^, grain N and grain Si accumulation (Appendix A, Figure 1).

### 2.2. Leaf Chlorophyll Index, Inorganic (N-NH_4_^+^ and N-NO_3_^−^), Total N and Si Accumulation in Maize Plants

Silicon application provided 4.6% greater LCI compared to the absence of Si application (Appendix A, Table 1). In addition, N application (low, medium and high N rates) provided greater LCI relative to control treatments (absence of side-dressing N fertilization) (Appendix A, Table 2).

Interaction between Si application and N levels significantly affected N-NO_3_^−^ and N-NH_4_^+^ accumulation in the root tissue (Appendix A, Figure 1a,b). The low (50 kg N ha^−1^) and medium (100 kg N ha^−1^) N application levels together with Si application provided greater root N-NO_3_^−^ and N-NH_4_^+^ accumulation (Appendix A, Figure 1a,b). Additionally, in the absence of N fertilization, Si application increased root N-NH_4_^+^ accumulation compared to plants from the treatments that did not receive Si (Appendix A, Figure 1b). Similarly, as verified for LCI, increasing N levels provided greater N-NO_3_^−^, N-NH_4_^+^, total N and Si accumulation (Table 2, Figure 1). In addition, root N-NH_4_^+^ and Si accumulation were greater with Si application relative to the absence of Si application, regardless of the year of cultivation (Appendix A, Table 1, Figure 1c). Root Si accumulation showed an increase of 33% in Si-applied plots compared to the plots without Si application (Table 1). Nitrate, N-NH_4_^+^ and total N accumulation in roots were greater in the first year than the second (Appendix A, Table 3, Figure 1c).

Interaction between Si amendment application and N levels: Lowercase letters indicate significant differences between Si application, and uppercase letters indicate significant differences between N levels according Tukey test (*p ≤* 0.05). Interaction between Si application and years: Lowercase letters indicate significant differences between Si application, and uppercase letters indicate significant differences between years according to the Tukey test (*p ≤* 0.05). Error bars indicate standard deviations (*n* = 4). −Si and +Si refer to the absence and presence of Si amendment application, respectively; C0, NLow, NMed and NHigh refer to absence, 50, 100 and 200 kg N ha^−1^ applied in side-dressing respectively; 1 and 2 refer to the first and second cropping season (2015/16 and 2016/17).

The application of Si positively influenced N-NO_3_^−^, N-NH_4_^+^ and Si accumulation in shoot compared to plots where no Si was applied (Appendix A, Table 1). Shoot Si accumulation increased by 19% in Si-applied plots compared to the plots that did not receive Si application (Table 1). The interaction between Si application and N levels significantly affected shoot total N accumulation (Appendix A, Figure 1d). Shoot total N accumulation was greater with Si application in both the presence and absence of N application. For example, shoot total N was 24% greater with Si application in the absence of N fertilization (+Si = 224 vs. −Si = 181 kg ha^−1^) (Figure 1d). Meanwhile, shoot total N was 26, 15 and 16% greater with Si application combined with 50, 100 and 200 kg N ha^−1^, respectively (50N: +Si = 280 vs. −Si = 222 kg ha^−1^; 100N: +Si = 308 vs. −Si = 267 kg ha^−1^; 200N: +Si = 320 vs. −Si = 277 kg ha^−1^) (Figure 1d). Shoot N-NO_3_^−^, N-NH_4_^+^ and total N accumulation tended to be greater with medium and high N application levels relative to the control and low N levels (Table 2, Figure 1). Meanwhile, shoot Si accumulation was greater with N application (regardless of N level—low, medium or high) when compared to the control (Table 2). Shoot N-NH_4_^+^, total N and Si accumulation were greater in first than second year (Table 3).

Regarding total N and Si accumulation in the grain, Si application increased grain N and Si accumulation in both the presence and absence of N application (Figure 1e,g). Grain N was 21% greater with Si application in the absence of N fertilization (+Si = 121 vs. −Si = 100 kg ha^−1^) (Figure 1e). However, grain N was 15, 16 and 13% greater with Si application combined with 50, 100 and 200 kg N ha^−1^ compared with plots that did not receive Si, respectively (50N: +Si = 148 vs. −Si = 129 kg ha^−1^; 100N: +Si = 162 vs. −Si = 140 kg ha^−1^; 200N: +Si = 173 vs. −Si = 153 kg ha^−1^) (Figure 1e). The accumulation of Si in grain was 62% greater with Si application in the absence of N fertilization (+Si = 0.97 vs. −Si = 0.60 kg ha^−1^) (Figure 1g). Similarly, grain Si was 41, 18 and 40% greater with the application of Si combined with 50, 100 and 200 kg N ha^−1^, respectively (50N: +Si = 1.1 vs. −Si = 0.78 kg ha^−1^; 100N: +Si = 1.3 vs. −Si = 1.1 kg ha^−1^; 200N: +Si = 1.4 vs. −Si = 1.0 kg ha^−1^) (Figure 1g). In general, grain N and Si accumulation tended to be greater with medium and high N levels and in the first year compared with the second year regardless of Si application (Figure 1f,h).

### 2.3. Maize Biomass, Grain Yield and Agronomic Efficiency

Root biomass increased with Si application in the absence of N fertilization and with a low N level relative to those plants from treatments that did not receive Si (Appendix A, Figure 1i). In contrast, high N application in plots that did not receive Si showed increased root biomass compared with plots that received Si (Appendix A, Figure 1i). In addition, root biomass was greater in the second year than in the first year (Appendix A, Table 3).

The application of Si resulted in 5.1% greater shoot biomass along with 5.2% greater grain yield compared to the absence of Si application (Table 1). Both shoot biomass and grain yield tended to be greater with the application of medium and high N levels compared to the control and low N levels (Table 2). Additionally, shoot biomass and grain yield were greater in first year than in the second year (Table 3).

Agronomic efficiency was increased by 73 and 44% with Si application under low and medium N levels compared to plots that did not receive Si application (50N: +Si = 30.2 vs. −Si =17.5 kg grains kg N applied^−1^; 100N: +Si = 22.1 vs. −Si =15.4 kg grains kg N applied^−1^) (Figure 1j). Agronomic efficiency tended to decrease with increasing N levels, regardless of Si application (Figure 1j).

### 2.4. Pearson’s Correlation among Si Accumulation in Maize Plant and the Other Evaluated Parameters

Overall, Pearson’s correlation was positive among Si accumulation in maize plant (root, shoot and grain) and biomass components (root, shoot biomass and grain yield), agronomic efficiency, LCI, N accumulation in plant (N accumulation in shoot, root and grain) and inorganic N accumulation in plant (N-NO_3_^−^ and N-NH_4_^+^ accumulation in shoot and root) (Figure 2). In contrast, Si grain accumulation was negatively correlated with total N and N-NO_3_^−^ accumulation in the root tissue (Figure 2).

### 2.5. Summary of Statistical Analysis: Wheat

Silicon application significantly affected root N-NO_3_^−^, root N-NH_4_^+^, shoot N-NO_3_^−^, shoot N-NH_4_^+^, shoot N, shoot Si, grain N and grain Si accumulation, shoot biomass and grain yield (Appendix A, Table 4). Nitrogen application significantly affected root N-NO_3_^−^, root N-NH_4_^+^, shoot N-NO_3_^−^, shoot N-NH_4_^+^, shoot N, shoot Si accumulation and grain N accumulation, shoot biomass and grain yield (Appendix A, Table 5). Years of study significantly affected root N-NO_3_^−^, root N-NH_4_^+^, root N, shoot N-NO_3_^−^, shoot N-NH_4_^+^, shoot Si, grain N and grain Si accumulation; shoot biomass and grain yield (Appendix A, Table 6).

The interaction between silicon application and N levels affected LCI, root Si accumulation, root biomass and agronomic efficiency (Appendix A, Figure 3). The interaction between silicon application and years of study significantly affected LCI, root Si accumulation and root biomass (Appendix A, Figure 3).

### 2.6. Leaf Chlorophyll Index, Inorganic (N-NH_4_^+^ and N-NO_3_^−^), Total N and Si Accumulation in Wheat Plants

Leaf chlorophyll index was 12 and 8.9% greater with Si application when coupled with low and medium N levels compared to without Si application (50N: +Si = 58 vs. −Si = 52; 100N: +Si = 61 vs. −Si = 56) (Appendix B, Figure 3a). Additionally, higher LCI was observed with medium and high N levels compared to control and low N levels regardless of Si application (Appendix B, Figure 3a). In addition, LCI was greater in the first year than in the second year (Figure 3a).

Interaction between Si amendment application and N levels: Lowercase letters indicate significant differences between Si application, and uppercase letters indicate significant differences between N levels according Tukey test (*p ≤* 0.05). Interaction between Si application and years: Lowercase letters indicate significant differences between Si application, and uppercase letters indicate significant differences between years according Tukey test (*p ≤* 0.05). Error bars indicate standard deviations (*n* = 4). −Si and +Si refers to absence and presence of Si amendment application, respectively; C0, NLow, NMed and NHigh refers to absence, 50, 100 and 200 kg N ha^−^^1^ applied in side-dressing respectively; 1 and 2 refers to the first and second cropping season (2015/16 and 2016/17).

Nitrate and N-NH_4_^+^ accumulation in the root were greater with Si application relative to the absence of Si and tended to increase with high N application compared to control, low and medium N levels (Appendix B, Table 4 and Table 5). Root inorganic (N-NO_3_^−^, N-NH_4_^+^) and total N were greater in the first year than in the second year (Appendix B, Table 6). Interaction between Si application and N levels significantly affected root Si accumulation (Appendix B, Figure 3c). The root Si accumulation was 53% greater with Si application in the absence of N fertilization (+Si = 2.6 vs. −Si = 1.7 kg ha^−1^) (Figure 3c). Additionally, root Si accumulation was 28, 20 and 38% greater with Si application combined with 50, 100 and 200 kg N ha^−1^, respectively (50N: +Si = 2.3 vs. −Si = 1.8 kg ha^−1^; 100N: +Si = 2.4 vs.−Si = 2.0 kg ha^−1^; 200N: +Si = 2.9 vs. −Si = 2.1 kg ha^−1^) (Figure 3c).

Nitrate, N-NH_4_^+^, total N and Si accumulation were increased with Si application (Appendix B, Table 4). We verified an increase of 10 and 45% in total N and Si accumulation in plots that received Si application compared to plots that did not receive Si application (Table 4). Inorganic N (N-NO_3_^−^ and N-NH_4_^+^), total N and Si accumulation in shoot tissue tended to be greater with medium and high N application levels compared to the control and low N levels (Appendix B, Table 5). In addition, N-NO_3_^−^, N-NH_4_^+^ and Si accumulation were greater in the first year than in the second year (Appendix B, Table 6).

Nitrogen and Si accumulation in the grain were 17 and 37% greater with Si application compared to the absence of Si (Appendix B, Table 4). Both grain N and Si accumulation were greater in the first year than in the second year (Appendix B, Table 6). Similarly, N accumulation in the shoot and grain increased with medium and high N application levels compared to the control and low N levels (Appendix B, Table 5).

### 2.7. Wheat Biomass, Grain Yield and Agronomic Efficiency

Root biomass was greater in plot receiving Si combined with low and high N levels application relative to plots that did not receive Si application (Appendix B, Figure 3e). Medium and high N levels provided greater root biomass coupled or not with Si application (Appendix B, Figure 3e). The positive effect of Si application was more evident in the second year due to the greater root biomass observed in plots receiving Si compared with plot that did not receive Si application (Appendix B, Figure 3f).

Shoot biomass and grain yield were increased by 13 and 7.6% with Si application compared to the plots that did not receive Si (Appendix B, Table 4). Shoot biomass was greater with medium and high N application levels compared to the control and low N levels (Appendix B, Table 5). However, grain yield was found to be greater with medium N level compared to the control, low and high N levels (Appendix B, Table 5).

Agronomic efficiency was 59 and 85% greater with Si application combined with low and high N levels compared to the absence of Si application (50N: +Si = 19.4 vs. −Si = 12.2 kg grain kg N applied^−1^ 100N: +Si = 5.0 vs. −Si = 2.7 kg grain kg N applied^−1^) (Appendix B, Figure 3g). The Si application along with increasing N levels (low > medium > high) tended to reduce agronomic efficiency (Appendix B, Figure 3g). Whereas, in the absence of Si application, the high N application level decreased agronomic efficiency compared to low and medium N levels (Figure 3g).

### 2.8. Pearson’s Correlation among Si Accumulation in Wheat Plants and the Other Evaluated Parameters

Similarly as verified in maize, overall, Pearson’s correlation was positive among Si accumulation in wheat plant (root, shoot and grain) and biomass components (root, shoot biomass and grain yield), LCI, N accumulation in plant (N accumulation in shoot, root and grain) and inorganic N accumulation in plant (N-NO_3_^−^ and N-NH_4_^+^ accumulation in shoot and root) (Figure 4). In contrast, Si root accumulation was negatively correlated with LCI (Figure 4). Meanwhile, Si grain accumulation was negatively correlated with shoot biomass and N root accumulation (Figure 4).

## 3. Discussion

Our results showed that Ca and Mg silicate was an effective source of Si which increased Si accumulation in the shoots and roots of field-grown maize and wheat crops. Most of the absorbed Si was accumulated in the shoot tissue, where it is deposited within the leaf epidermis through plant transpiration flux. The Si inside plant epidermis was condensed into a polymerized silica gel (SiO_2_ nH_2_O) known as a phytolith that is immobile and developed a protective structural layer in the plant cell walls [36,37,38]. Interestingly, we verified that grain Si accumulation in maize and wheat also increased with silicate application. Some plant species can take up and translocate large quantities of Si in the aboveground tissue due to specific Si transporters [39]. For example, Ma et al. [40] reported that the transport of Si from soil to the rice panicles is mediated by different transporters. According to these authors, Lsi1, belonging to a NIP group of the aquaporin family, is responsible for the uptake of Si from soil into the root cells in both dicots and monocots although its expression patterns and cellular localization differ with plant species. The subsequent transport of Si out of the root cells towards the stele is medicated by an active efflux transporter, Lsi2 [40]. Silicon in the xylem sap is presented in the form of monosilicic acid and is unloaded by Lsi6, a homolog of Lsi1 in rice [40]. Lsi6 is also involved in the inter-vascular transfer of Si at the node, which is necessary for preferential Si distribution to the panicles [40].

Silicon-applied treatments showed a total Si accumulation (shoot, root and grain) of 43 kg Si ha^−1^ for maize and 36 kg Si ha^−1^ for the wheat crop. Some previous studies reported that sugarcane (*Saccharum officinarum* L.) has the potential to take up the largest amount of silicon (300–700 kg of Si ha^−1^), followed by rice (150–300 kg of Si ha^−1^), and wheat (30–150 kg of Si ha^−1^) [41]. On an average, plants can absorb anywhere from 30 to 300 kg Si ha^−1^ [38,42]. Such values of absorbed silicon cannot be fully explained by passive absorption because the upper 0.20 m soil layer contains only an average of 0.1–1.6 kg Si ha^−1^ as monosilicic acid [41]. The above high Si accumulation observed in maize plants indicated that this crop could be considered one of the most efficient Si accumulators in *Poaceae*, after sugarcane and rice.

Leaf chlorophyll index (LCI) was benefited by Si application in maize. The Si application also benefited LCI in wheat, however the positive response was more evident in the first year than in the second wheat crop season when coupled with low and medium N levels. Silicon fertilization can benefit the foliar architecture of plants by improving erectness of leaves which leads to greater light interception, reducing self-shading and lodging, postponing senescence and improving photosynthesis [39,43]. Additionally, Si application had greater influence on the agronomic efficiency of the N fertilizer applied to the maize and wheat crops. Specifically, under low and medium N levels, agronomic efficiency was increased by 73 and 44% with Si application in maize, respectively. Similarly, under low and high N levels, agronomic efficiency was increased by 58 and 85% with Si application in wheat. Silicon application can enhance primary metabolism by decreasing the transpiration rate [44] and improving photosynthesis [45,46] and nutrient uptake [47]. Thereby, the greater Si uptake and LCI verified in our study together with enhanced agronomic efficiency of N-applied levels due to silicate application positively influenced the inorganic N accumulation in maize and wheat root and shoot, and grain N accumulation, leading to a greater shoot biomass and grain yield. The positive Pearson’s correlation along with Si accumulation in maize and wheat plants, LCI, and N accumulation in plant and biomass components support this hypothesis (Figure 2 and Figure 4).

The results suggested that Si application increases N agronomic efficiency and grain yield, so presumably less N would need to be applied when used in combination with Si. Based on the verified maize and wheat grain yield, linear interpolation was accessed (the equations were presented as Appendix C). It would be possible to reduce N application from 180–185 to 100 kg N ha^−1^ (for maize) and from 195–200 to 100 kg N ha^−1^ (for wheat) when silicate was applied, with similar grain yield (maize: −Si 180–185N = 10223–10264 kg ha^−1^ vs. +Si 100N = 10252 kg ha^−1^; wheat: −Si 195–200N = 4385–4405 kg ha^−1^ vs. +Si 100N = 4392 kg ha^−1^) (Appendix C). Positive responses to Si application, mainly under abiotic and biotic environments, were reported elsewhere for different plant species [48,49,50,51,52]. Here, we have focused on the positive effects of Si application on N agronomic efficiency under harsh field growing conditions, such as drought and high temperatures which are common under tropical agriculture conditions. In our study, although total rainfall was relatively well distributed during maize growing seasons, high temperatures were observed during the field trial (maximum and minimum temperature above 35 °C and 20 °C in most of the cases, respectively) (Figure 5). Regarding wheat crop, supplementary irrigation was performed, although total rainfall was poor and not well distributed during the wheat growing seasons (Figure 5). In addition, high temperatures were observed during the field trial (maximum and minimum temperature above 30 °C and 15 °C in most of the cases, respectively). Thus, plant growth response to Si application under tropical conditions would be more noticeable, as verified in our study, especially for Si accumulator crops such as maize and wheat, as we discussed above.

Although verifying a positive residual effect of Si application, our study indicated that the influence of residual Si-amendment source was less evident in each successive crop season. For example, shoot and grain Si accumulation was greater in the first than second maize cropping season. In addition, the shoot and grain N accumulation, and the shoot biomass and grain yield followed the same trend as observed for the shoot and grain Si accumulation. Our study observed the same behavior in wheat crop, with a reduced shoot and grain Si accumulation followed by reduced LCI, grain N accumulation, shoot biomass and grain yield. We could expect this behavior once Si uptakes by maize-wheat crops are higher and coupled with no other nutrient application would reduce Si availability along successive crop seasons. Thus, adequate Si replenishment along continuous successive cropping seasons are required to avoid this element depletion and impaired cereal growth and yield.

## 4. Materials and Methods

### 4.1. Site Description

The study was carried out under field conditions in Selvíria (Savannah region—Brazilian Cerrado), state of Mato Grosso do Sul, Brazil (20°22′ S and 51°22′ W, 335 m above sea level (a.s.l.), during the 2015 to 2017 growing seasons. The soil was classified as Rhodic Haplustox (Clayey Oxisol) according to the Soil Survey Staff [53]. Twenty random soil samples were collected from the entire experimental site with a soil core sample type cup auger (0.10 m × 0.40 m—cup diameter and length, respectively) from 0.00–0.20 m depth. The samples were mixed and a random subsample from each depth was collected, air-dried, sieved (2-mm), and stored at ambient temperature until analyses. The subsample was used for the determination of soil chemical attributes and granulometry prior to soil amendment sources application on October 9th, 2015. Soil chemical attributes were analysed according to Raij et al. [54] and granulometry followed Teixeira et al. [55]. Total N was determined by the semi-micro Kjeldahl method [56]. Silicon was determined after extraction in Ca chloride (0.01 mol L^−1^) according to the methodology of Korndörfer et al. [57] (Table 7).

The experimental area had been cultivated with annual leguminous and cereal crops for over 30 years. In addition, the area has been under no-tillage for the last 15 years. The crop sequence prior to the field trial was maize (2014), soybean (*Glycine max* (L.) Merr.) (2014/15), maize (2015), maize (2015/16), wheat (2016), maize (2016/17) and wheat (2017). The 2015 maize crop received the application of 180 kg N ha^−1^ on the entire experimental site. From maize 2015/16 crop until wheat 2017 crop, specific rates of Si and N were performed in each plot. The maximum, average, and minimum temperatures and rainfall observed during the field trial are reported in Figure 5.

### 4.2. Experimental Design and Treatments

The experimental design was a randomized complete block design (RCBD), with four replicates arranged in a 2 × 4 full factorial scheme. The treatments comprised two soil amendment sources: (1) Si application (+Si) treatment: Ca and Mg silicate (one single product—10.5% Si, 25% CaO, and 6% MgO, solubility of 0.095 g L^−1^) with an effective neutralizing power (ENP) of 88%, and (2) absence of Si application (−Si): dolomitic limestone (28% CaO and 20% MgO, solubility of 0.014 g L^−1^) with an ENP of 80%; and four N levels, applied in side-dress N application: (1) control (0 kg N ha^−1^), (2) low (50 kg N ha^−1^), (3) medium (100 kg N ha^−1^), and (4) high (200 kg N ha^−1^) levels. These levels were selected based on previous studies with N management in maize and wheat crops under Brazilian tropical Savannah conditions [35]. The applied N source was urea (45% of N) at six leaves completely unfolded (at V6 phenological stage) of maize and wheat tillering—decimal growth stage GS21 [58].

Wheat planting took place on the same plots of the preceding crop (maize) and the same plots were used for the entire duration of the study. The experimental plots were six maize rows of five meters spaced at a distance of 0.45 m, when maize was cultivated (13.5 m^2^) and twelve wheat rows of five meters spaced at a distance of 0.17 m, when wheat was cultivated (10.2 m^2^). The useful central area of the plot was considered as 10 m^2^ (5 m × 2 m) and was used as reference point to plant maize and wheat.

Soil amendment sources (Si treatments) were broadcast applied at the rate of 1.76 Mg ha^−1^ for the silicate (equivalent to 185 kg Si ha^−1^) and 1.94 Mg ha^−1^ for the limestone 30 days before planting maize (2015/16) in a one-time, single application, without reapplication in subsequent years. No incorporation was performed as the area was under no-tillage. The amount of lime applied was based on the initial soil analysis and the amount needed to increase the base saturation (V) to 80% based on the following Equation (1):(1)$$\mathrm{SAN}=\frac{\mathrm{CEC}\;\left(V2-V1\right)}{10\times \mathrm{ENP}}$$
where SAN = Soil amendment needed, in Mg ha ^1^; CEC = cation exchange capacity; V2 = base saturation to be achieved; V1 = current base saturation and ENP = effective neutralizing power.

### 4.3. Maize and Wheat Management

A simple maize hybrid DOW 2B710 was sown in both cultivations (2015/16 and 2016/17) at a density of 7.3 viable seeds m^−2^ with a no-till drill. At maize sowing, basal fertilization application was performed for all treatments with 375 kg ha^−1^ of granular fertilizer 08-28-16 (N-P_2_O_5_-K_2_O) in the sowing furrows based on the soil analysis and maize crop requirements [59]. As we mentioned before, N side-dressing (N application levels) was manually applied at V6 phenological stage to evenly distribute the fertilizer on soil surface without incorporation. Maize was cultivated during 13 November 2015 to 15 March 2016 (harvested 116 days after emergence) and 11 November 2016 to 21 March 2017 (harvested 125 days after emergence) in 2015/16 and 2016/17 cropping season, respectively. Weeds were controlled using pre and post-emergence herbicides, and insects were controlled using best management practices in maize. When necessary, maize crop was irrigated with supplementary irrigation through a center pivot sprinkling system (water depth of 14 mm).

The wheat cultivar CD 1104 was sown in both cultivated years (2016 and 2017) at a density of 412 viable seeds m^−2^ with a no-till drill. Additionally, basal fertilization application was performed at sowing for all treatments with 275 kg ha^−1^ of the granular fertilizer 08-28-16 (N-P_2_O_5_-K_2_O) in the sowing furrows based on the soil analysis and crop requirements [59]. Nitrogen application in side-dressing (N application levels) was performed manually to evenly distribute the fertilizer on the soil surface without incorporation during wheat tillering. The crop was irrigated with supplementary irrigation using a center pivot sprinkling system (water depth of 14 mm) based on the crop moisture requirement. The crop was cultivated during 3 May 2016 to 8 September 2016 (harvested 120 days after emergence) and 10 May 2017 to 12 September 2017 (harvested 117 days after emergence). Weeds were controlled using pre- and post-emergence herbicides, and insects were controlled using best management practices for wheat.

### 4.4. Samplings and Analysis

Five maize plants and wheat plants from an area of 0.17 m^2^ (0.17 m—width of the row × 1.0 m) were harvested at ground level during flowering for shoot collection. At the same time, a side trench of approximately 0.50 m depth was dug for root collection and then washed with deionized water. In addition, leaf chlorophyll index (for maize and wheat), was determined indirectly by collecting readings on 20 maize ear leaves and 30 wheat flag leaves and when plants reached the flowering stage, using a portable nondestructive ClorofiLOG^®^ meter (model CFL-1030, Falker, Porto Alegre, Brazil). The shoots and roots were dried for 90 h in a forced-air oven at 60 °C. Then, the shoots and roots biomass (kg ha^−1^) were weighed. The Si concentration was determined following Silva [60] and total N analysis followed the methodology proposed by Malavolta et al. [61]. Silicon were quantified in an Ultraviolet—visible spectroscopy spectrophotometer (UV-VIS—Model Varian Cary-50, Varian, Victoria, Australia). Nitrogen was distilled by semi-micro Kjeldahl method and determined by titration with HCl 0.05 N. The inorganic N concentration (N-NO_3_^−^ and N-NH_4_^+^) in plant tissues was determined following Tedesco et al. [62]. Briefly, 1 g of plant tissue was extracted with 1 mol KCl L^−1^ (ratio of 1:15, plant tissue: solution, *w*/*v*), distilled with MgO (N-NH_4_^+^) and Devarda’s alloy and titrated with 2.5 mmol H_2_SO_4_ L^−1^ (N-NO_3_^−^). The accumulated Si, total N, N-NH_4_^+^ and N-NO_3_^−^ in shoot and root were obtained by the product of Si, total N and inorganic N concentrations in tissue and produced biomass. Grain yield was determined by spike collection from the useful lines of each maize and wheat plots. After mechanical harvest, the grains were weighted (kg ha^−1^ with 13% moisture content in wet basis). Grain Si and N accumulation were determined following the same procedures as shoot and root Si and N accumulation. Agronomic efficiency of applied N-fertilizer (AE) was calculated following Fageria et al. [63] methodology presented in Equation (2).
AE (kg grain kg N applied^−1^) = (grain yield at Nx − grain yield at N0)÷N level applied(2)
where Nx is the N level applied and N0 is the control treatment (without N applied in side-dressing).

### 4.5. Statistical Analysis

All data were initially tested for Levene’s homoscedasticity test (*p* ≤ 0.05) and normality using Shapiro and Wilk test, which showed the data to be normally distributed (*W* ≥ 0.90). Data were submitted to analysis of variance (*F* test) using repeated measures (cropping seasons (maize and wheat, separately) as the repeated variable) and a compound symmetry model for the covariance parameters. When a significant main effect or interaction was observed by the *F* test (*p* ≤ 0.05), the Tukey test (*p* ≤ 0.05) was used for comparison of means of Si application, N levels, cropping seasons and their interactions using the ExpDes package. To identify dependent variables directly related to Si accumulation in maize and wheat plants, a Pearson correlation analysis (*p* ≤ 0.05) was performed. To create a heatmap, the corrplot package was used, using the “cor” and “cor.mtest” functions to calculate the coefficients and *p*-value matrices. Asterisks were added to the heatmap cells to the identification of significant correlations. All the statistical analysis was performed using R software [64].

## 5. Conclusions

Silicon amendment application enhanced leaf chlorophyll index, agronomic efficiency and N-uptake in maize and wheat plants, benefiting shoot and root development and leading to a higher grain yield (an increase of 5.2 and 7.6%, respectively). We verified that it would be possible to reduce N fertilization in maize from 180-185 to 100 kg N ha^−1^ while maintaining similar grain yield with Si application. Additionally, when Si application was performed, it would be possible to reduce N fertilization in wheat from 195-200 to 100 kg N ha^−1^. Therefore, in view of a high probability of a positive response to maize-wheat cropping sequence even associated with different N application levels, the Si amendment application could be a key technology for improving plant-soil N-management, especially in Si accumulator crops and leading to a more sustainable cereal production under tropical conditions.

## Figures and Tables

**Figure 1 plants-10-01329-f001:**
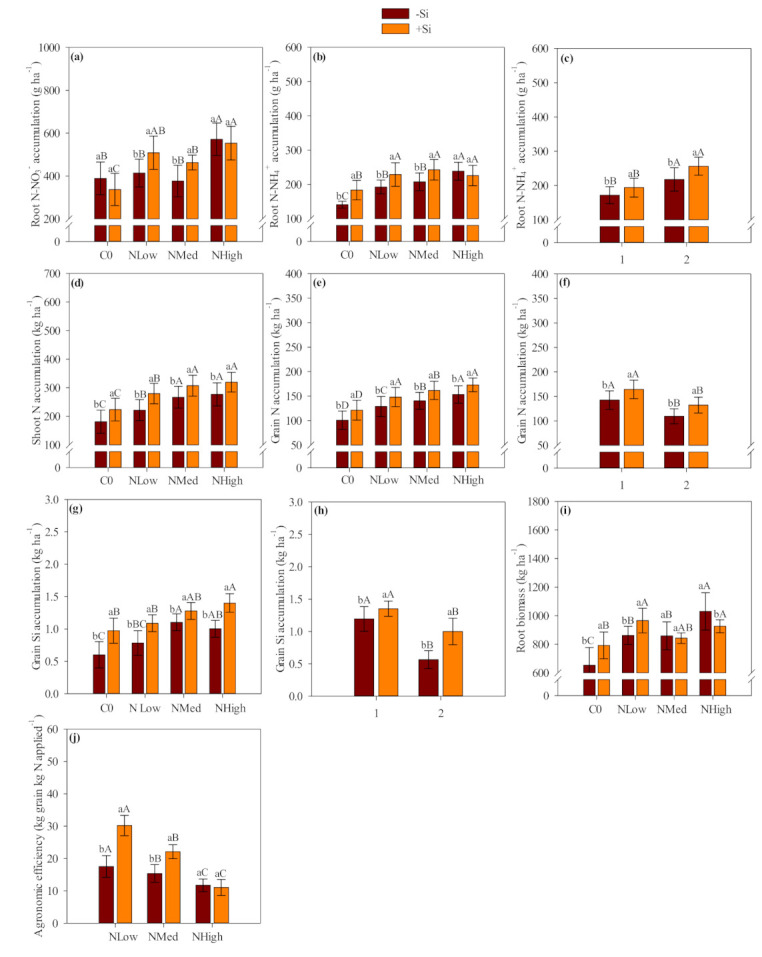
Maize root nitrate (N-NO_3_^−^) accumulation (**a**), root ammonium (N-NH_4_^+^) accumulation (**b**,**c**), shoot N accumulation (**d**), grain N accumulation (**e**,**f**), grain Si accumulation (**g**,**h**), root biomass (**i**) and agronomic efficiency (**j**) affected by the interaction between Si amendment application and N levels or by the interaction between Si amendment application and years of study.

**Figure 2 plants-10-01329-f002:**
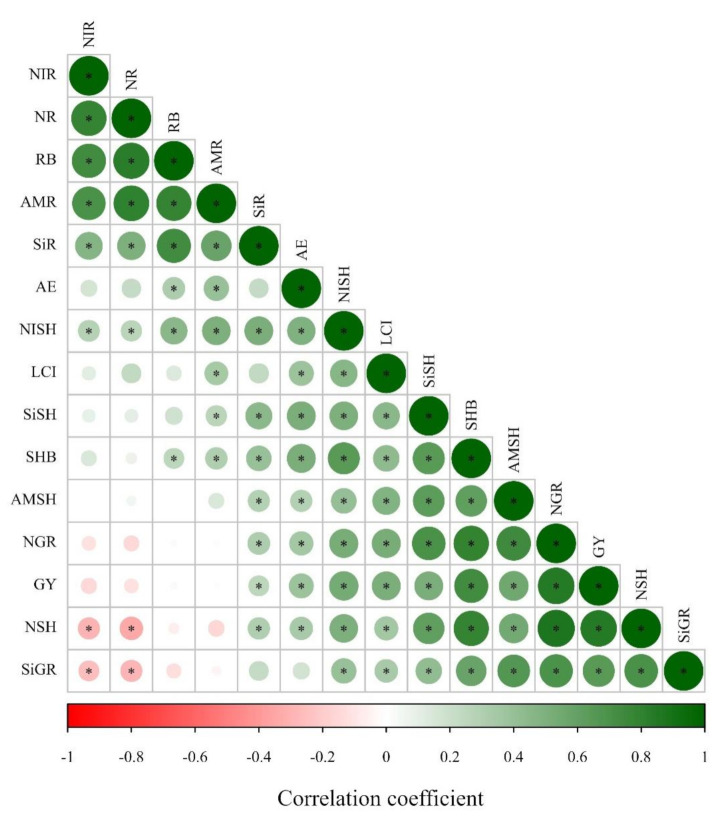
Heatmap showing the Pearson’s correlation among the analyzed parameters in maize plants in response to Si application and N levels. * indicates significant relationship (*p* ≤ 0.05). Abbreviation: NIR = N-NO_3_^−^ accumulation in root, NR = total N accumulation in root, RB = root biomass, AMR = N-NH_4_^+^ accumulation in root, SiR = Si accumulation in root, AE = agronomic efficiency, NISH = N-NO_3_^−^ accumulation in shoot, LCI = leaf chlorophyll index, SiSH = Si accumulation in shoot, SHB = shoot biomass, AMSH = N-NH_4_^+^ accumulation in shoot, NGR = total N accumulation in grain, GY = grain yield, NSH = total N accumulation in shoot, SiGR = Si accumulation in grain.

**Figure 3 plants-10-01329-f003:**
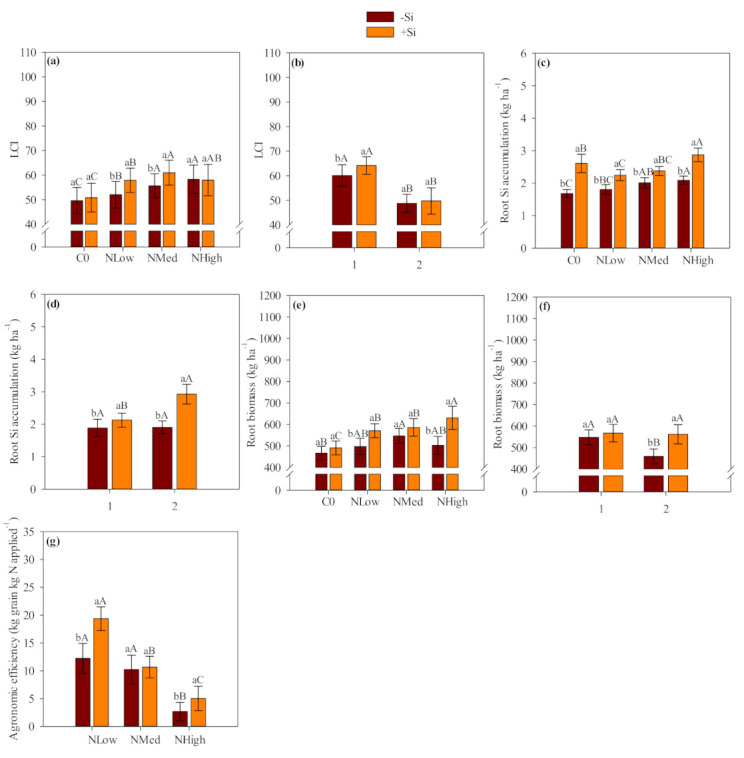
Wheat leaf chlorophyll index (LCI) (**a**,**b**), root Si accumulation (**c**,**d**), root biomass (**e**,**f**) and agronomic efficiency (**g**) affected by the interaction between Si amendment application and N levels or by the interaction between Si amendment application and years of study.

**Figure 4 plants-10-01329-f004:**
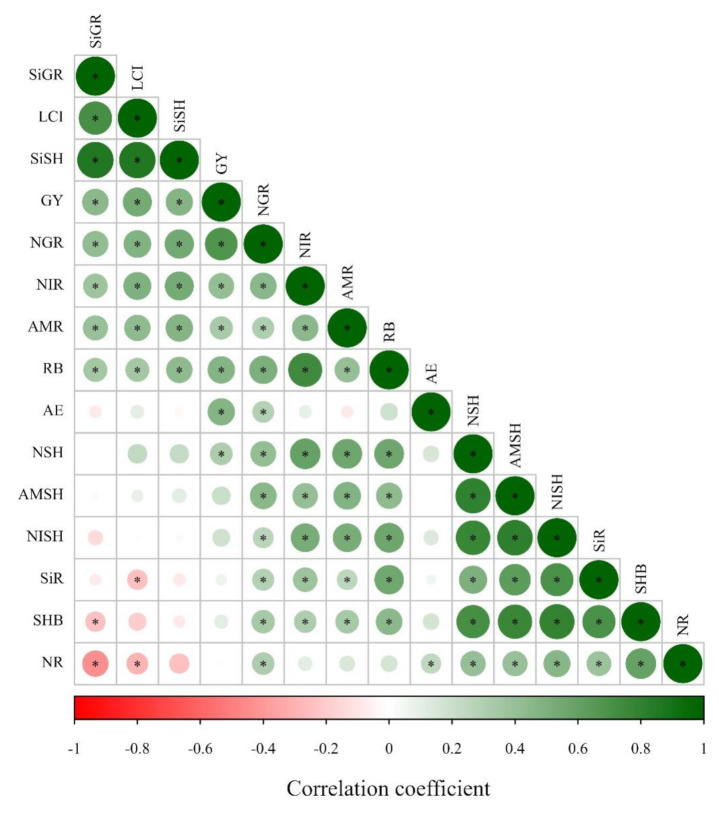
Heatmap showing the Pearson’s correlation among the analyzed parameters in wheat plants in response to Si application and N levels. * indicates significant relationship (*p* ≤ 0.05). Abbreviation: NIR = N-NO_3_^−^ accumulation in root, NR = total N accumulation in root, RB = root biomass, AMR = N-NH_4_^+^ accumulation in root, SiR = Si accumulation in root, AE = agronomic efficiency, NISH = N-NO_3_^−^ accumulation in shoot, LCI = leaf chlorophyll index, SiSH = Si accumulation in shoot, SHB = shoot biomass, AMSH = N-NH_4_^+^ accumulation in shoot, NGR = total N accumulation in grain, GY = grain yield, NSH = total N accumulation in shoot, SiGR = Si accumulation in grain.

**Figure 5 plants-10-01329-f005:**
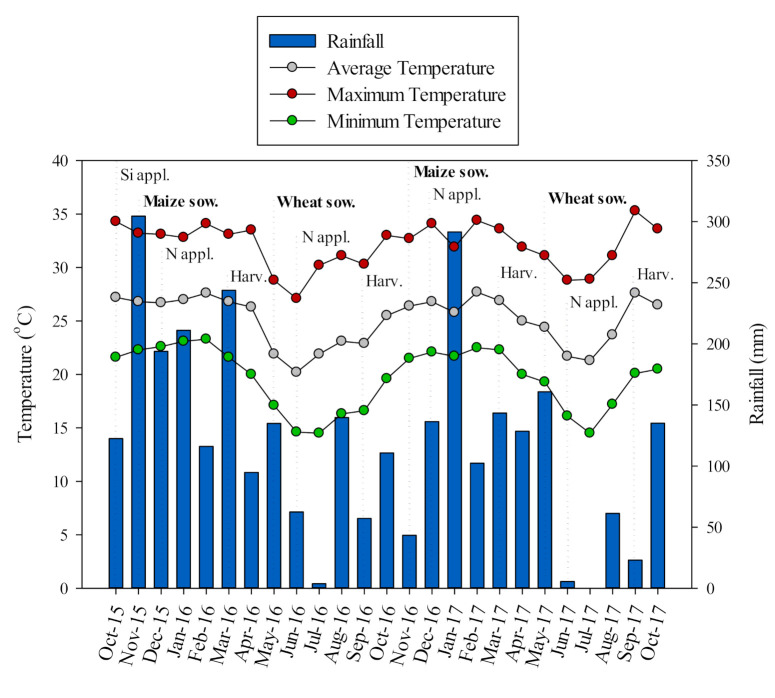
Rainfall, and temperatures (minimum, average and maximum) obtained from the weather station located in the Education and Research Farm of College of Engineering of Ilha Solteira/São Paulo State University (FEIS/UNESP) during the period of October 2015 to October 2017. Vertical dotted lines represent the Si-amendment source application, maize and wheat sowing, N application in side-dressing and harvest dates, respectively.

**Table 1 plants-10-01329-t001:** Maize leaf chlorophyll index (LCI); root Si accumulation; shoot nitrate (N-NO_3_^−^); shoot ammonium (N-NH_4_^+^); shoot Si accumulation; shoot biomass and grain yield as a function of Si amendment application.

Sources	LCI	Root Si	Shoot N-NO_3_^−^	Shoot N-NH_4_^+^
		(kg ha^−1^)	---------------- (g ha^−1^) ----------------	
−Si	65.9 b ± 2.9	7.30 b ± 0.58	2157 b ± 187	2423 b ± 168
+Si	68.9 a ± 3.0	9.71 a ± 0.41	2408 a ± 198	2715 a ± 171
**Sources**	**Shoot Si**	**Shoot Biomass**	**Grain Yield**	
	------------------------------------- (kg ha^−1^) -------------------------------------			
−Si	27.2 b ± 1.5	11713 b ± 526	9230 b ± 434	
+Si	32.2 a ± 1.8	12296 a ± 539	9711 a± 390	

Different letters indicate significant differences between treatments according to Tukey test (*p* ≤ 0.05). ± indicate standard deviations (*n* = 4). −Si and +Si refers to absence and presence of Si amendment application.

**Table 2 plants-10-01329-t002:** Maize leaf chlorophyll index (LCI); root N; root Si accumulation; shoot nitrate (N-NO_3_^−^); shoot ammonium (N-NH_4_^+^); shoot Si accumulation; shoot biomass and grain yield as a function of N levels.

**N Rates**	**LCI**	**Root N**	**Root Si**	**Shoot N-NO_3_^−^**
		(kg ha^−1^)		(g ha^−1^)
Control (C 0N)	63.4 b± 2.8	4.00 c ± 0.55	6.84 c ± 0.60	1837 c ± 216
N Low	67.3 a ± 2.8	5.17 ab ± 0.55	8.35 b± 0.64	2262 b ± 177
N Medium	69.3 a ± 2.7	4.89 b ± 0.50	8.33 b ± 0.49	2419 ab ± 194
N High	69.4 a ± 3.0	5.76 a ± 0.51	10.47 a ± 0.68	2610 a ± 247
**N Rates**	**Shoot N-NH_4_^+^**	**Shoot Si**	**Shoot biomass**	**Grain yield**
	(g ha^−1^)	(kg ha^−1^)		
Control (C 0N)	2182 b ± 149	24.6 b ± 2.24	9482 c ± 296	8415 c ± 363
N Low	2524 a ± 163	31.4 a ± 2.15	12077 b ± 505	9233 b ± 358
N Medium	2767 a ± 206	31.5 a ± 1.80	13181 a ± 505	9913 a ± 364
N High	2801 a ± 140	31.3 a ± 2.24	13277 a ± 577	10320 a ± 361

Different letters indicate significant differences between treatments according to Tukey test (*p* ≤ 0.05). ± indicate standard deviations (*n* = 4). Ctl, Low, Med and High refers to absence, 50, 100 and 200 kg N ha^−1^ applied in side-dressing respectively.

**Table 3 plants-10-01329-t003:** Maize root nitrate (N-NO_3_^−^); root N accumulation; shoot ammonium (N-NH_4_^+^); shoot N; shoot Si accumulation; root biomass; shoot biomass and grain yield as a function of years of study.

**Year**	**Root N-NO_3_^−^**	**Root N**	**Shoot N-NH_4_^+^**	**Shoot N**
	(g ha^−1^)	(kg ha^−1^)	(g ha^−1^)	(kg ha^−1^)
1	357 b ± 75	4.09 b ± 0.47	2712 a ± 178	316 a ± 32
2	545 a ± 86	5.82 a ± 0.49	2424 b ± 177	208 b ± 32
**Year**	**Shoot Si**	**Root Biomass**	**Shoot Biomass**	**Grain Yield**
	(kg ha^−1^)			
1	31.2 a ± 1.92	795 b ± 123	12727 a ± 552	10309 a ± 336
2	28.1 b ± 1.88	936 a ± 138	11281 b ± 484	8632 b ± 433

Different letters indicate significant differences between treatments according to Tukey test (*p* ≤ 0.05). ±indicate standard deviations (*n* = 4). 1 and 2 refer to the first and second cropping season (2015/16 and 2016/17).

**Table 4 plants-10-01329-t004:** Wheat root nitrate (N-NO_3_^−^); root ammonium (N-NH_4_^+^) accumulation; shoot N-NO_3_^−^; shoot N-NH_4_^+^; shoot N; shoot Si accumulation; grain N; grain Si accumulation; shoot biomass and grain yield as a function of Si amendment application.

Sources	Root N-NO_3_^−^	Root N-NH_4_^+^	Shoot N-NO_3_^−^	Shoot N-NH_4_^+^	Shoot N
	(g ha^−1^)				(kg ha^−1^)
−Si	121 b ± 7.93	83.7 b ± 4.59	1336 b ± 143	1116 b ± 130	127 b ± 11.1
+Si	143 a ± 8.85	99.2 a ± 5.52	1629 a ± 146	1319 a ± 116	141 a ± 10.5
**Sources**	**Shoot Si**	**Grain N**	**Grain Si**	**Shoot Biomass**	**Grain Yield**
	(kg ha^−1^)				
−Si	20.44 b ± 2.49	102 b ± 7.52	3.01 b ± 0.32	4841 b ± 218	3925 b ± 154
+Si	29.00 a ± 3.72	118 a ± 7.22	4.07 a ± 0.55	5491 a ± 238	4223 a ± 130

Different letters indicate significant differences between treatments according to Tukey test (*p* ≤ 0.05). ± indicate standard deviations (*n* = 4). −Si and +Si refers to absence and presence of Si amendment application.

**Table 5 plants-10-01329-t005:** Wheat root nitrate (N-NO_3_^−^); root ammonium (N-NH_4_^+^) accumulation; shoot N-NO_3_^−^; shoot N-NH_4_^+^; shoot N; shoot Si accumulation; grain N accumulation; shoot biomass and grain yield as a function of N levels.

N Rates	Root N-NO_3_^−^	Root N-NH_4_^+^	Shoot N-NO_3_^−^	Shoot N-NH_4_^+^	Shoot N
	(g ha^−1^)				(kg ha^−1^)
Control (C 0N)	104 c ± 7.85	84.2 b ± 5.12	1218 c ± 119	1002 c ± 119	109 c ± 11.4
N Low	122 bc ± 8.27	87.5 ab ± 5.32	1475 b ± 142	1109 bc ± 106	122 b ± 12.9
N Medium	145 ab ± 8.32	92.3 ab ± 5.28	1649 ab ± 119	1353 b ± 104	143 b ± 10.1
N High	156 a ± 8.67	101 a ± 4.99	1789 a ± 124	1426 a ± 102	148 a ± 11.1
**N Rates**	**Shoot Si**	**Grain N**	**Shoot Biomass**	**Grain Yield**	
	(kg ha^−1^)				
Control (C 0N)	19.4 b ± 1.81	97.9 b ± 7.78	4617 b ± 243	3540 b ± 242	
N Low	24.2 a ± 2.03	106 ab ± 7.62	5198 ab ± 241	4174 a ± 190	
N Medium	26.5 a ± 1.69	119 a ± 8.56	5367 ab ± 248	4427 a ± 205	
N High	28.6 a ± 1.96	115 a ± 7.35	5581 a ± 254	4154 a ± 189	

Different letters indicate significant differences between treatments according to Tukey test (*p* ≤ 0.05). ± indicate standard deviations (*n* = 4). Ctl, Low, Med and High refers to absence, 50, 100 and 200 kg N ha^−^^1^ applied in side-dressing respectively.

**Table 6 plants-10-01329-t006:** Wheat root nitrate (N-NO_3_^−^); root ammonium (N-NH_4_^+^); root N accumulation; shoot N-NO_3_^−^; shoot N-NH_4_^+^; shoot Si accumulation; grain N; grain Si accumulation; shoot biomass and grain yield as a function of years of study.

Year	Root N-NO_3_^−^	Root N-NH_4_^+^	Root N	Shoot N-NO_3_^−^	Shoot N-NH_4_^+^
	(g ha^−1^)		(kg ha^−1^)	(g ha^−1^)	
1	146 a ± 7.14	97.8 a ± 5.81	5.86 a ± 0.59	1580 a ± 111	1275 a ± 107
2	117 b ± 8.24	85.1 b ± 5.23	4.45 b ± 0.54	1385 b ± 154	1160 b ± 108
**Year**	**Shoot Si**	**Grain N**	**Grain Si**	**Shoot Biomass**	**Grain Yield**
	(kg ha^−1^)				
1	37.4 a ± 2.41	116 a ± 8.27	4.83 a ± 0.50	5512 a ± 244	4273 a ± 243
2	11.9 b ± 2.1	103 b ± 7.53	2.25 b ± 0.65	4820 b ± 274	3875 b ± 201

Different letters indicate significant differences between treatments according to Tukey test (*p* ≤ 0.05). ± indicate standard deviations (*n* = 4). 1 and 2 refers to the first and second cropping season (2016 and 2017).

**Table 7 plants-10-01329-t007:** Soil chemical attributes and granulometry in 0–0.20 m depth before Si amendment source application in 2015.

Soil Chemical Attributes	Unit	0–0.20 m Layer
Total N	g kg^−1^	1.04
P (resin)	mg kg^−1^	19
S (SO_4_)	mg kg^−1^	10
Organic matter	g kg^−1^	21
pH (CaCl_2_)		5.0
K	mmol_c_ kg^−1^	2.1
Ca	mmol_c_ kg^−1^	19.0
Mg	mmol_c_ kg^−1^	13.0
H+Al	mmol_c_ kg^−1^	28.0
Al	mmol_c_ kg^−1^	1.0
B (hot water)	mg kg^−1^	0.17
Cu (DTPA)	mg kg^−1^	3.1
Fe (DTPA)	mg kg^−1^	20.0
Mn (DTPA)	mg kg^−1^	27.2
Zn (DTPA)	mg kg^−1^	0.8
Cation exchange capacity (pH 7.0)	mmol_c_ kg^−1^	62.1
Base saturation	%	55
**Granulometry**		**0–0.20 m Layer**
Sand	g kg^−^^1^	471
Silt	g kg^−^^1^	90
Clay	g kg^−^^1^	439

*n* = 20, DTPA = diethylenetriaminepentaacetic acid.

## Data Availability

All data generated or analyzed during this study are included in this published article.

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
