# Peer review of "Silicon Amendment Enhances Agronomic Efficiency of Nitrogen Fertilization in Maize and Wheat Crops under Tropical Conditions"

_plants, 2021, doi:10.3390/plants10071329_

Round 1
Reviewer 1 Report
This study raises a topic of current interest related to the importance of silicon in agriculture and particularly its role on plant nutrition and N-use efficiency. Especially since the study was carried out in an open field conditions, while the effectiveness of silicon on plants is often provided via greenhouse works
However, the authors did not establish an adequate format to present the results and meet the stated aims. The way in which the results/graphs are presented make them not understandable. Each analyzed parameter is presented differently from the other without any homogeneity, making difficult the interpretation of the results. Also, putting results on interactions between Si amendment application and N levels (/or year) in the same graphs is confusing and not relevant.
It’s not clear if authors provide silicon as a single product (Ca-Mg silicate), or as a mix of two silicate products (Ca-silicate and Mg-silicate), and some basal informations (like solubility) on such product are needed. Indeed, authors indicated that silicate are 6-7 times more soluble than limestone (lines 67-68). If that is the case, it would be important to take into consideration the provision of Ca and Mg as nutrients for the plants during the cropping seasons. While dolomitic limestone serves to balance the base saturation without taking into account if Ca and Mg are available (or not) for plants. Therefore the conclusions of the paper should be reconsidered, because we should not attribute all the effects observed on growth and N-use efficiency to silicon only, but also to Ca and Mg.
For the reasons given below, I submit this manuscript to a major revision
Author Response
Dear editor,
We are sending attached the point-by-point response to reviewers.
All suggested revisions were performed.
Thank you.
The authors

Reviewer 2 Report
In the manuscript “Silicon Amendment Enhances Agronomic Efficiency of Nitrogen Fertilization in Maize and Wheat Crops under Tropical Conditions« authors Fernando Shintate Galindo, Paulo Humberto Pagliari, Willian Lima Rodrigues , Guilherme Carlos Fernandes, Eduardo Henrique Marcandalli Boleta, José Mateus Kondo Santini, Arshad Jalal, Salatiér Buzetti, José Lavres and Marcelo Carvalho Minhoto Teixeira Filho, investigate the combined effects of Si-amendment source and N levels application on leaf chlorophyll index, inorganic and total N and Si content in maize and wheat plants, plant biomass, grain yield and agronomic efficiency of individual treatments.
Abstract is OK.
Key words Are OK.
Introduction
Is informative, concise, with the appropriate literature cited.
Materials and Methods
Clear and well explained
Results
L 151 – 152 Regarding total N and Si accumulation in the grain. Si application increased grain N and Si accumulation in both the presence and absence of N application (Figs. 2a and 2b). Write comma instead of full stop
Discussion
Is informative, concise, with the appropriate literature cited.
L 310-311You verified that grain Si accumulation in maize and wheat also increased with silicate application. Explain possible transport route please.
Specific comments
The idea of MS is interesting and results clearly presented. I would like to highlight the clear and comprehensive work of the authors. The methods employed are explained and the results consistent with the purposes. Discussion is sound and relevant.
Reference 32 is not written in the same style as others.
My suggestions: minor revision
Author Response

(The authors gave the same response as above.)

Reviewer 3 Report
The topic of this paper is interesting and the experimental design is appropriate. However, the presentation of the results is uncomfortable. For example, Figures 1 and 2 are complex, badly designed and unintelligible.
The authors need to significantly improve the quality of these figures to convince the reader that their results are relevant and innovative.
In the introduction section, some references are missing. For example, a recent study has shown that Si supply to Brassica napus grown under field conditions leads to better fertilizer utilization and could reduce nitrogen application rates.
As it stands, the manuscript is not acceptable for publication and needs to be significantly improved.
Author Response

(The authors gave the same response as above.)

Round 2
Reviewer 1 Report
I thank the authors for making changes to the manuscript, but this remains insufficient for a good understanding of the results. Here are some of interrogations regarding data processing and presentation :
Table 1 : the presented results are for which cropping season ? and which nitrogen level ? why we could not find the results on root N-NO3- ? While we find them in figure 1a. why root biomass are not presented ? but this parameter was presented in figure 1i.
Results presented in Table 2 are not relevant, also, it is not clear if it is the results of season 1 or 2 and which Si-condition
Why we do not find the shoot biomass within figure 1 ? some measured parameters are presented under the different N-regimes and Si-treatment but for which cropping season ?
Figure 1 : Why only results on root-NH4, grain N and grain Si are presented depending on the cropping season ? and why they are not presented in regards of the N-level as for the other parameters ?
Results presented in Table 2 are not relevant. Why authors want to compare results of season 1 and 2 ? If they provide relevant conclusion then authors must specify which N- and Si-treatment they used.
All together make the format of the result not coherent and still confusing. Also, regarding the data processing, nothing is mentioned in the material and method section
Protocol and experimental design is well established. For each cropping season and each modality, there is two harvesting stages : flowering and crop maturity.
At flowering they measures : LCI, root and shoot biomass, root and shoot Si content, root and shoot N content, root and shoot N-NO3 content and root and shoot N-NH4 content.
At maturity they measures : yield, grain Si and N-content and AE
It could be judicious to present separately the results of the first and second season. And also separate the results of the first (flowering) and the second (maturity) harvest. For each measured parameter, highlight the effect of Si and N-level on the same model as the figure 1a for example.
For each crop and crop season, It could be interesting to present the results on yield as a curve response to nitrogen with or without Si.
Conclusion is a copy-past of a part of the abstract, please rephrase one or the other
For the reasons given below, I submit this manuscript to a major revision
Author Response
Dear Editor,
We are submitting attached the point-by-point response letter to reviewer 1.
Best regards,
The authors

Reviewer 3 Report
In despite of a significant improvement of the design of figuress ( especially, figs 1 and 2), presentation of results remains unclear and confusive.
For example, in new tables (1, 2 etc.) clustered unclear data previously presented in the old fig 1 (or 2 etc.) Obviously, there is again a serouis lack of informations concerning the data presented in these new tables: years?, N treatments etc?
The same remark can be made about figures 1 and 2. For example fig 1e and 1f (and this is only an example applicable to other data) presenting the yield, what is the year of experimentation for figure 1e? What is the N treatment in figure 1f?
The acceptance of this manuscript depends on a clear and limpid presentation of the results through the various figures and tables.
As minor revisions:
In tables: write medium instead of average
In figure and tables:
-Add N to Low, med and High
-Replace Ctl by C (0N)
Introduction including new references is now adequate. However, thank you to respect the following numbering for the citation of the new references.
Author Response
Dear Editor,
We are submitting attached the point-by-point response letter to reviewer 3.
Best regards,
The authors

Round 3
Reviewer 1 Report
I thank the authors for making changes to the manuscript and provides an accurate explanation of the statistical method they used which clearly make sense. The summary of the statistical analysis in results section is useful
Reviewer 3 Report
accepted-
Minir revisions:
Write C(0N) Low N, Med N and High N instead of (C0) NLow, N medium and Nhigh